# Binary architecture of the Na$_v$1.2-β2 signaling complex

**Samir Das[1†], John Gilchrist[2†], Frank Bosmans[2,3*], Filip Van Petegem[1*]**

[1]Department of Biochemistry and Molecular Biology, Life Sciences Institute, University of British Columbia, Vancouver, Canada; [2]Department of Physiology, Johns Hopkins University School of Medicine, Baltimore, United States; [3]Solomon H Snyder Department of Neuroscience, Johns Hopkins University School of Medicine, Baltimore, United States

**Abstract** To investigate the mechanisms by which β-subunits influence Na$_v$ channel function, we solved the crystal structure of the β2 extracellular domain at 1.35Å. We combined these data with known bacterial Na$_v$ channel structural insights and novel functional studies to determine the interactions of specific residues in β2 with Na$_v$1.2. We identified a flexible loop formed by $^{72}$Cys and $^{75}$Cys, a unique feature among the four β-subunit isoforms. Moreover, we found that $^{55}$Cys helps to determine the influence of β2 on Na$_v$1.2 toxin susceptibility. Further mutagenesis combined with the use of spider toxins reveals that $^{55}$Cys forms a disulfide bond with $^{910}$Cys in the Na$_v$1.2 domain II pore loop, thereby suggesting a 1:1 stoichiometry. Our results also provide clues as to which disulfide bonds are formed between adjacent Na$_v$1.2 $^{912/918}$Cys residues. The concepts emerging from this work will help to form a model reflecting the β-subunit location in a Na$_v$ channel complex.

**\*For correspondence:**
frankbosmans@jhmi.edu (FB); filip.
vanpetegem@gmail.com (FVP)

†These authors contributed
equally to this work

**Competing interests:** The
authors declare that no
competing interests exist.

**Reviewing editor:** David E
Clapham, Howard Hughes
Medical Institute, Boston
Children's Hospital, United
States

## Introduction

Voltage-gated sodium (Na$_v$) channels are part of membrane-embedded signaling complexes that initiate the rising phase of action potentials, a crucial event in generating and propagating electrical signals throughout the human body (*Hille, 2001*; *Catterall, 2012*). As key components of these protein assemblies, β-subunits (1) modify Na$_v$ channel-gating properties; (2) regulate channel trafficking and localization to distinct surface compartments; and (3) influence channel oligomerization (*Calhoun and Isom, 2014*; *Namadurai et al., 2015*). Moreover, β-subunits can alter the toxin pharmacology of Na$_v$ channels (*Gilchrist et al., 2014*), a concept that has been exploited to detect their presence in heterologous expression systems or native tissues (*Wilson et al., 2011*; *Zhang et al., 2013*; *Wilson et al., 2015*; *Gilchrist et al., 2013*). Structurally, β-subunits are single-transmembrane segment glycoproteins with a short cytoplasmic C-terminal tail and a large V-type immunoglobulin (Ig) extracellular domain that may participate in homophilic and heterophilic interactions, cell adhesion, and cell migration (*Calhoun and Isom, 2014*; *Namadurai et al., 2015*; *Brackenbury et al., 2008*). Although all β-subunits belong to the Ig family, recent atomic resolution information for the β3 and β4 extracellular domain revealed substantial differences in their 3D structure (*Gilchrist et al., 2013*; *Namadurai et al., 2014*). Given their distinct features and functional roles, it has now become clear that each β-subunit structure should be obtained and assessed separately. Of the four known β-subunits and their splice variants (β1–4; gene names *Scn1b-Scn4b*) (*Isom et al., 1992*; *Isom et al., 1995a*; *Morgan et al., 2000*; *Patino, 2011*; *Yu, 2003*), β2 and β4 form a disulfide bond with an unidentified Cys within particular Na$_v$ channel isoforms. In contrast, non-covalent interactions underlie β1 and β3 association with Na$_v$ channels as well as other members of the voltage-gated ion channel family (*Calhoun and Isom, 2014*; *Namadurai et al., 2015*; *Marionneau, 2012*; *Nguyen et al.,*

**eLife digest** Our bodies run on electricity. The brain, heart and some other organs depend on small electrical signals that are generated by ions moving through specialized protein complexes that sit in the membrane surrounding a cell. One of these channels is a 'sodium channel', through which positively charged sodium ions move. Tiny changes in the structure of the sodium channel can cause severe conditions such as epilepsy and heart arrhythmias, so it is crucial that we know how it works

Sodium channels consist of different protein building blocks (called α and β) and it was not known exactly how these come together to form the full channel complex. However, previous studies hinted at which parts of the β building block make contact with the α protein.

Now, Das, Gilchrist et al. have been able to visualize the three-dimensional structure of the β building block of the sodium channel in extremely high detail by using a technique called X-ray crystallography. The level of detail in the structure also allowed the amino acids that make up the β building block to be identified.

Das, Gilchrist et al. then altered some of the amino acids in the sodium channel, and treated frog cells containing the mutant channel with a spider toxin that binds between the α and β building blocks. This revealed the location and identity of the exact contact points between the proteins. In the future, a full three-dimensional structure showing the α and β subunits bound together would yield invaluable information on how they cooperate to form the sodium channel complex and give insights into mutations that cause cardiac arrhythmias and epilepsy.

*2012*; *Deschenes et al., 2008*). Aberrant behavior of the ubiquitously expressed β2 and β4 subunits has been linked to disorders such as long-QT syndrome, atrial fibrillation, sudden infant death syndrome, and epilepsy, possibly through dysregulation of the $Na_v$ channel signaling complex (*Li et al., 2013*; *Medeiros-Domingo et al., 2007*; *Tan et al., 2010*; *Baum et al., 2014*; *Watanabe et al., 2009*). Moreover, β2 has been implicated in neurodegenerative disorders and neuropathic pain and is therefore of potential interest for developing novel therapeutic strategies (*O'Malley et al., 2009*; *Lopez-Santiago, 2006*). Finally, β2 is targeted by secretase enzymes, an observation that suggests a potential contribution to Alzheimer's disease (*Gersbacher et al., 2010*; *Kim et al., 2005*).

Despite accruing evidence supporting their clinical relevance, fundamental questions on the causal relationship between β2 mutations and disorders remain unanswered. In contrast, auxiliary proteins are the topic of herculean research efforts in other fields where their role as vital contributors to cellular function or in forming drug receptor sites is well established (*Copits and Swanson, 2012*; *Gee et al., 1996*; *Milstein and Nicoll, 2008*; *Dolphin, 2012*). To begin to appreciate β2 function and lay the foundations for constructing an interacting model with $Na_v$ channels, we first need to define the mechanism by which β2 regulates $Na_v$ channel function. In particular, knowledge of the relative orientation of both partners will help to assess whether a mutation modifies channel function or influences complex assembly. To this end, we solved the crystal structure of the extracellular β2 domain at 1.35Å and found a [55]Cys-containing binding region that modulates $Na_v$1.2 toxin susceptibility by β2. Next, we exploited a combination of $Na_v$ channel mutagenesis, biochemistry, and β2-induced alterations in spider toxin pharmacology to uncover a disulfide bond between [55]Cys and a partnering Cys located in the domain II S5-S6 pore loop of $Na_v$1.2. Remarkably, $Na_v$1.5 and close relatives $Na_v$1.8/$Na_v$1.9 do not possess a corresponding Cys, which may explain a lack of β2 effect on $Na_v$1.5 toxin pharmacology. In concert with the available structural information on bacterial $Na_v$ channels (*Payandeh et al., 2012*; *2011*; *Shaya et al., 2014*; *Zhang et al., 2012*), our results provide conceptual insights into the location of the β2-subunit in the $Na_v$ channel-signaling complex.

## Results

### Crystal structure of the human β2 extracellular region

To begin to understand the functional relationship between human (h)β2 and h$Na_v$ channels at an atomic level, we solved the crystal structure of the extracellular hβ2 domain at a resolution of 1.35Å

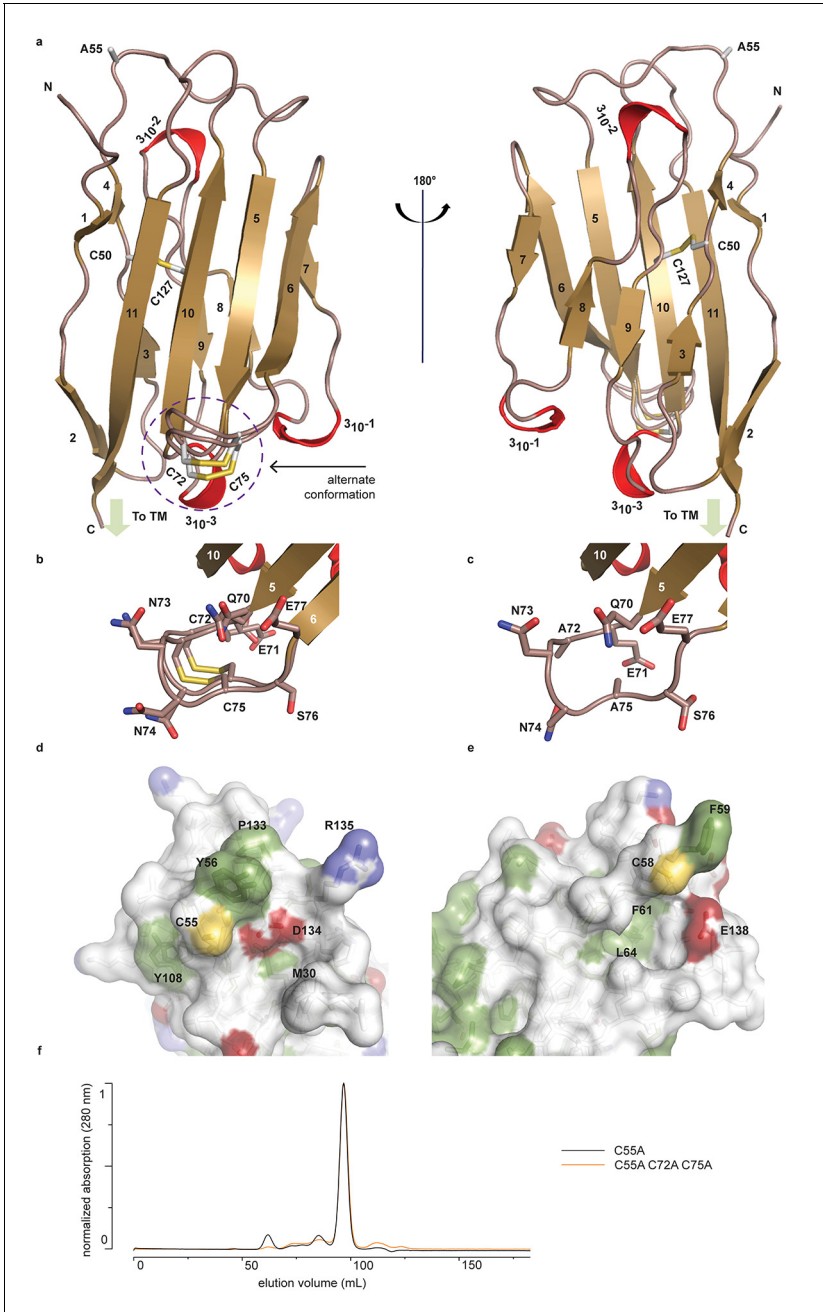

**Figure 1.** Crystal structure of hβ2. (a) Cartoon representation of the hβ2 (C55A) extracellular domain crystal structure, showing β strands in gold and $3_{10}$ helices in red. Cysteine side chains, as well as the [55]Ala residue, are shown in stick representation. Positions of N-terminus (N) and C-terminus (C) are indicated. The loop containing the [72]Cys- [75]Cys disulfide bond is modeled in a dual conformation. (b) Detail of the dual conformation of the loop, showing all side chains in stick conformation. (c) Detail of the same loop in the C72A/C75A (C55A) mutant, shown from the same viewpoint as in panel (b). (d,e) Comparison of the surfaces of hβ2 (d) and hβ4 (e) surrounding the reactive cysteines ([55]Cys and [58]Cys, respectively). Side chains of hydrophobic residues are shown in green, negatively charged carboxyl groups in red, and positively charged amino and guanidinium groups in blue. The position of the cysteine (which has been mutated to alanine to allow crystallization) is shown in yellow. (f) Size exclusion chromatograms (Preparative Superdex200) for hβ2 C55A and hβ2 C55/72/75A, which both elute as monomeric species.

The following source data is available for figure 1:

**Source data 1.** X-ray data collection and refinement statistics.

(*Figure 1a*, *Figure 1—source data 1*). The construct encompasses residues 30–153 (*Figure 2*), and contains a single Cys mutation (C55A) to facilitate crystallization [The C55A mutation is located on the protein surface and therefore unlikely to affect the overall structure (*Figure 1a,d*).]. The hβ2 configuration displays an Ig-like fold, consisting of eleven β-strands and three $3_{10}$ helices. Other than the mutated Cys (C55A), this domain contains four additional cysteines that are arranged in two bonds. The first, intra-subunit bond, is strictly conserved among all four β-subunit isoforms and is mediated by $^{50}$Cys and $^{127}$Cys buried within the core where it links two opposing faces of the protein. The second intra-subunit bond is located within a loop that spans residues 70–77 and connects strand $β_5$ to $β_6$ via $^{72}$Cys - $^{75}$Cys (*Figure 1a–b*). This loop constitutes a unique feature of hβ2 since corresponding cysteines are absent in β1, β3, and β4. Remarkably, this region displays a dual conformation in the crystal structure which indicates a high degree of flexibility (*Figure 1b*).

Although the overall hβ2 extracellular domain exhibits a similar fold compared to that of the previously reported hβ3[11] and hβ4[9] structures, considerable differences are observed. Aside from the distinct $^{72}$Cys - $^{75}$Cys disulfide bond, there are extensive variances in loop lengths and β strands, indicating that hβ2 and hβ4 have diverged substantially (*Figure 2a,c*). Yet, at a protein sequence level, hβ2 is most closely related to hβ4 (26% identical; 48% conserved) (*Figure 2d,e*). Of note is that the N-terminal region of hβ2 is structured rather than disordered in hβ4, adding an additional short β strand ($β_1$) which interacts with the novel $β_4$ strand. The strand following the unique disulfide bond is shortened in hβ2 and two additional $3_{10}$ helices can be seen, whereas one that is present in hβ4, no longer exists in hβ2. A comparison with the hβ3 structure also shows substantial divergence in loop lengths at particular locations (*Figure 2b,c*). The hβ3 extracellular domain misses the additional N-terminal β-strand; instead, its N-terminus is anchored via a disulfide bridge between $^{26}$Cys and $^{48}$Cys, resulting in positional shifts in excess of 15Å.

We previously found that $^{58}$Cys on the surface of β4 is crucial in modulating Na$_v$1.2 susceptibility to toxins from spider and scorpion venom (*Gilchrist et al., 2013*). The hβ2 subunit possesses a corresponding Cys at position 55 that is located in a longer loop which may result in an altered spatial position of this residue when compared to hβ4 (*Figure 2a,c*, *Figure 1d,e*). In addition, the residues immediately N-terminal to $^{55}$Cys are stabilized through a β-strand interaction between strands $β_1$ and $β_4$, both of which are absent in hβ4. Taken together, these observations suggest that the environment of this key Cys may differ in both isoforms. To further examine the coupling of hβ2 to hNa$_v$1.2, we next determined the functional role of $^{55}$Cys as well as that of $^{72}$Cys and $^{75}$Cys.

## Mutating $^{55}$Cys in hβ2 restores hNa$_v$1.2 toxin susceptibility

Previously, we and others discovered that β-subunits can manipulate the pharmacological sensitivities of Na$_v$ channels (*Wilson et al., 2011*; *Gilchrist et al., 2013*; *Doeser et al., 2014*). In particular, β4 is capable of dramatically altering animal toxin binding to rat (r)Na$_v$1.2a (*Gilchrist et al., 2013*). For example, the spider toxin ProTx-II (*Middleton et al., 2002*) binds to voltage-sensing domains (VSDs) I, II, and IV in rNa$_v$1.2a (*Bosmans et al., 2008*), and is ~five-fold less potent when β4 is present. [An intriguing concept is that ProTx-II may also bind directly to β4 (*Gee et al., 1996*); however, our isothermal calorimetry experiments do not support this notion (*Figure 3—figure supplement 1*).] To examine whether this protective ability extends to hβ2, we expressed hNa$_v$1.2 in *Xenopus* oocytes and measured ProTx-II susceptibility without or in the presence of the β-subunit (*Figure 3— source data 1, Figure 3—figure supplement 2*). Similar to β4, we observe that the hβ2 subunit expresses abundantly and traffics to the membrane (*Figure 3d*) where it is able to reduce the degree of hNa$_v$1.2 current inhibition by ProTx-II. Specifically, 100 nM ProTx-II reduces hNa$_v$1.2 conductance to ~17% of peak whereas the current remaining in the presence of hβ2 is typically more than ~64% of peak conductance, thereby demonstrating a protective effect. Other gating parameters such as conductance-voltage (G–V) and channel availability relationships are unaffected (*Figure 3—source data 1*). Next, we sought to determine if $^{55}$Cys in hβ2 is involved in reducing hNa$_v$1.2 susceptibility to ProTx-II by mutating this residue to an Ala. Indeed, the C55A mutant traffics to the membrane and causes ProTx-II inhibition of hNa$_v$1.2 to resemble that of the wild-type (WT) channel without hβ2 present (*Figure 3c,d*, *Figure 3—figure supplement 2*). Although Ala wields a small side chain and is often employed in mutagenesis studies, we also replaced $^{55}$Cys with a Ser, as it most closely resembles Cys in terms of size and electric properties. Similar to WT hβ2 and C55A, the C55S mutation impaired neither channel expression nor surface trafficking (*Figure 3d*, *Figure 3—figure supplement 2*). Moreover, the extent of ProTx-II-induced hNa$_v$1.2 inhibition in the presence of

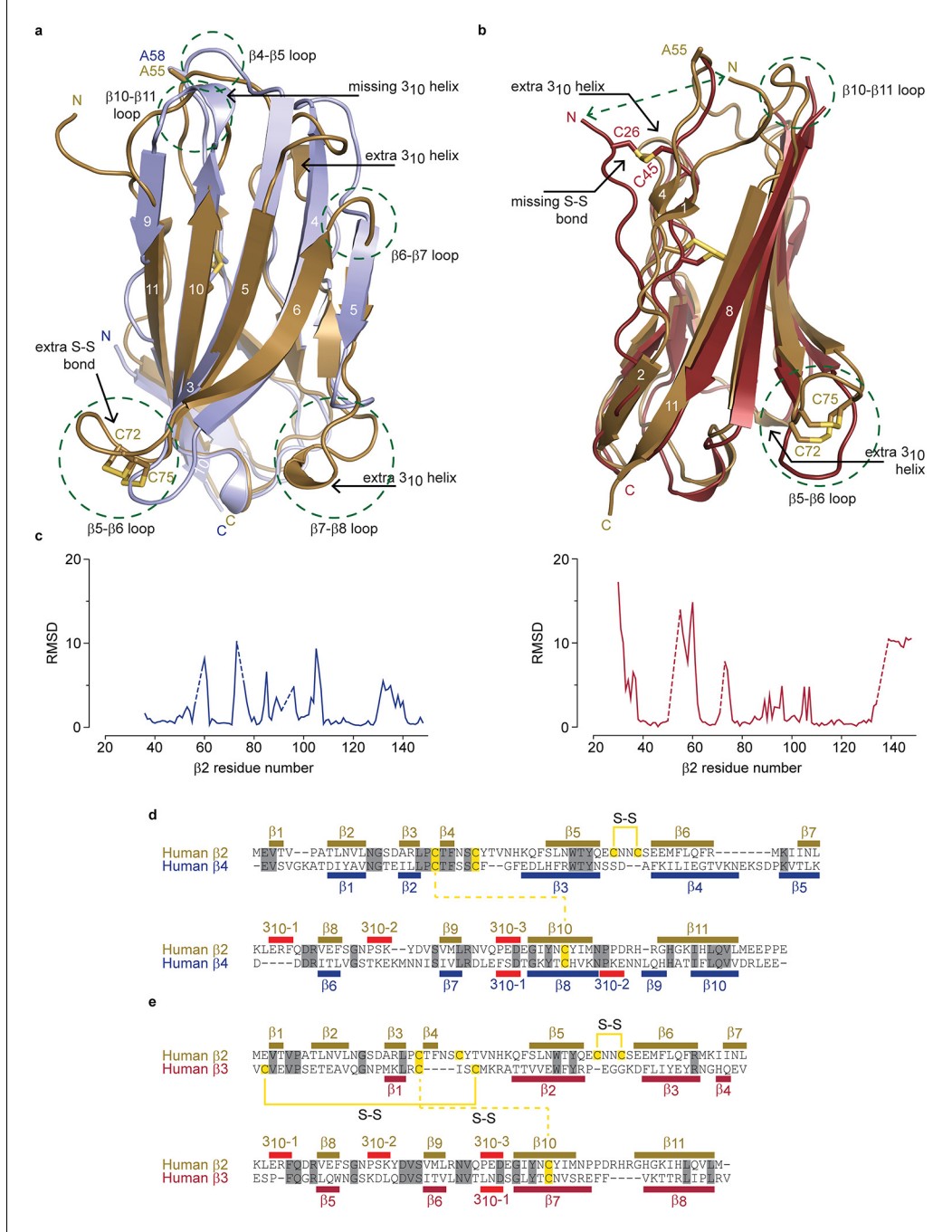

**Figure 2.** Structural comparison of hβ2 to hβ3 and hβ4. (a) Superposition of the crystal structures of hβ2 C55A (gold) and hβ4 C58A (blue) in cartoon representation. Cysteines or their equivalent residues (A55 and A58, respectively) are shown in sticks. The major differences are highlighted in the figure. (b) Superposition of crystal structures of hβ2 C55A (gold) and hβ3 (red). (c) Root-mean-square-deviation (RMSD) plots showing the RMSD values per residue for hβ4 C58A (left, blue plot) and hβ3 (right, red plot) relative to hβ2 after a superposition. The residue numbers below correspond to the hβ2 numbers. Sections for which there are no corresponding residues at indicated with dotted lines. (d,e) Shown are sequence alignments of the extracellular domains of (d) hβ2 versus hβ4 and (e) hβ2 versus hβ3. Conserved residues are highlighted in grey, and cysteines in yellow. Observed disulfide bonds are also labeled (S-S). Secondary structure elements found in the corresponding crystal structures are also shown.

The following figure supplement is available for figure 2:

**Figure supplement 1.** Amino acid sequence alignment of the β2 protein found in various organisms.

C55S is indistinguishable from that of the channel alone (*Figure 3c*, *Figure 3—source data 1*). Altogether, these results support the notion that hβ2 conveys ProTx-II protection to hNa$_v$1.2 via [55]Cys and may relate to previous work in which the loss of the covalent link between rβ2 and hNa$_v$1.1 disrupts the targeting of rβ2 to nodes of Ranvier and to the axon initial segment in hippocampal neurons (*Chen et al., 2012*).

In addition to [55]Cys, the hβ2 crystal structure reveals a unique motif bearing a protruding disulfide-stabilized loop formed by [72]Cys and [75]Cys (*Figure 1a–b*). Remarkably, this additional loop is highly conserved in almost all species that express β2, suggesting an evolutionary conserved contribution to function (see *Figure 2*, *Figure 2—figure supplement 1*). To determine if this loop regulates the gating or pharmacological influence of hβ2 on hNa$_v$1.2, we mutated [72]Cys and [75]Cys to Ala (C72A C75A) but found that it is functionally indistinguishable from WT hβ2 (*Figure 3c*, *Figure 3—figure supplement 2*). The C72A C75A mutant localizes to the oocyte membrane surface without or with hNa$_v$1.2 co-expression. Moreover, typical gating parameters and hNa$_v$1.2 inhibition by 100 nM ProTx-II in the presence of C72A C75A is similar to that observed for the channel when co-expressed with WT hβ2 (*Figure 3—source data 1*). The lack of effect of the C72A C75A mutant on hNa$_v$1.2 function suggests that this disulfide bond is not essential for folding, and that its disruption may not significantly affect the position or environment of [55]Cys. To verify this hypothesis, we produced recombinant hβ2 extracellular domain containing three Cys mutations: C55A, C72A, and C75A. Size exclusion chromatography demonstrates that the mutant produces monomeric protein, indicating that the bond between [72]Cys and [75]Cys is unessential for folding (*Figure 1f*). Furthermore, we obtained a crystal structure of the triple mutant at 1.85Å which overlays well onto the C55A structure (*Figure 1c*). The only significant difference is situated in the loop containing both cysteines, which now displays a single conformation. Although the spatial organization of this loop does not seem to impact the ability of hβ2 to modulate hNa$_v$1.2 gating or sensitivity to ProTx-II, this region may yet play a functional role in modulating other Na$_v$ channel isoforms.

## The S5-S6 loop in domain II of hNa$_v$1.2 contains an anchoring point for hβ2

Although previous work has postulated the involvement of the domain II (DII) S5-S6 pore loop as the region responsible for forming an inter-subunit disulfide bond between particular Na$_v$ channel isoforms and β2 or β4 (*Chen et al., 2012*; *Gajewiak et al., 2014*), the precise residue has remained elusive. To explore the possibility of an hβ2 anchoring point in this region, we individually replaced each of the three cysteines found here ([910]Cys, [912]Cys, and [918]Cys) with Ser. When expressed without or with WT hβ2, the C910S mutant exhibits the same degree of inhibition by 100 nM ProTx-II, indicating that the protective effect of hβ2 is lost and that [910]Cys is a critical residue for hβ2 binding (*Figure 4a–b*, *Figure 4—source data 1* and *Figure 4—figure supplement 1*). In contrast, the C918S mutant retains protection from 100 nM ProTx-II by the β-subunit. The C912S mutant displays a split toxin-sensitive population that is consistently observed throughout multiple oocyte batches. One fraction of experiments reveals hNa$_v$1.2 current inhibition as though no hβ2 is present whereas another displays protection against 100 nM ProTx-II, similar to the WT channel and C918S mutant (*Figure 4a*). To verify the presence of hβ2, oocytes were collected after recording and checked for expression by Western blot (see *Figure 4b*), thereby indicating that the observed loss-of-protection effects indeed relate to the hNa$_v$1.2 mutation.

Altogether, these results hint towards a possibility of mutation-induced shifts in intra-subunit disulfide bond formations, a plausible scenario given the close vicinity of [910]Cys, [912]Cys, and [918]Cys in hNa$_v$1.2. To evaluate this hypothesis, we constructed a set of double Cys to Ser mutations (C910S C912S, C910S C918S, and C912S C918S) and examined the effects on ProTx-II susceptibility without and in the presence of hβ2 (*Figure 4a–b*, *Figure 4—source data 1* and *Figure 4—figure supplement 1*). According to previous work with rNa$_v$1.2a, mutating [912]Cys or [918]Cys results in a bridge between [910]Cys and the remaining intact cysteine (*Zhang et al., 2015*). These experiments were carried out without a β2 subunit and with GVIIJ (*Gajewiak et al., 2014*), a unique μO§-conotoxin that directly competes for binding to [910]Cys; however, the rest of its binding site remains unexplored. In our hands, high concentrations of GVIIJ are needed to achieve an effect, and efficacy towards hNa$_v$1.2 is augmented in case of the C910S mutation (*Figure 4—source data 2*, *Figure 4—figure supplement 2*), two complicating factors that made us revert to ProTx-II which is more potent and has clearly delineated binding sites within Na$_v$1.2. Our experiments on these double mutants show a

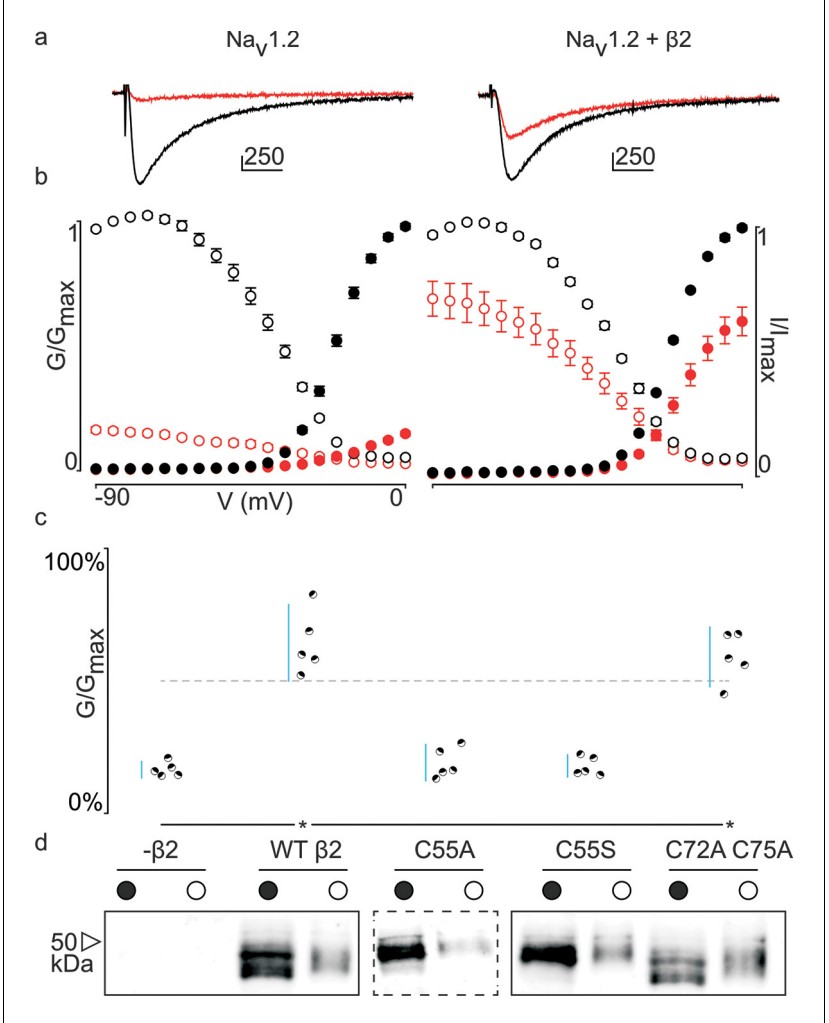

**Figure 3.** Effect of hβ2 on hNa$_v$1.2 toxin pharmacology. (**a**) Co-expression of hNa$_v$1.2 with hβ2 decreases the degree of inhibition by 100 nM ProTx-II. Left trace shows ProTx-II strongly inhibiting WT hNa$_v$1.2 whereas right trace displays attenuated inhibition in the presence of hβ2. Black trace is control condition without toxin, red is in the presence of ProTx-II. Traces depict a 50 ms depolarization to -15 mV from -90 mV. Scale bar is 10 ms on horizontal axis and given nA vertically. (**b**) Normalized conductance-voltage (G-V, filled circles) and steady-state inactivation (I-V, open circles) relationships for hNa$_v$1.2 with and without hβ2. Pre-toxin values are shown in black and post-toxin in red. Fit values can be found in **Figure 3—source data 1**. (**c**) Dot plot comparing hβ2 mutations by ability to prevent ProTx-II inhibition of hNa$_v$1.2. Black circles represent individual oocytes; vertical axis shows percent of inhibition by ProTx-II at peak conductance. Blue lines represent a 95% confidence interval. hβ2 mutations are presented underneath the horizontal axis and label the lanes below in (**d**). Statistical significance (p<0.01) is indicated by an asterisk. (**d**) Western blot against the C-terminal myc-tag of hβ2. No signal is seen in the negative control but is observed for the WT hβ2 and all mutants, both in whole cell (filled circle) and surface (open circle) fractions.

The following source data and figure supplements are available for figure 3:

**Source data 1.** Table providing values for fits of the data presented in **Figure 3** and **Figure 3—figure supplement 2**.

**Figure supplement 1.** ProTx-II does not bind directly to β4.

**Figure supplement 2.** G-V and SSI relationships for hβ2 mutants.

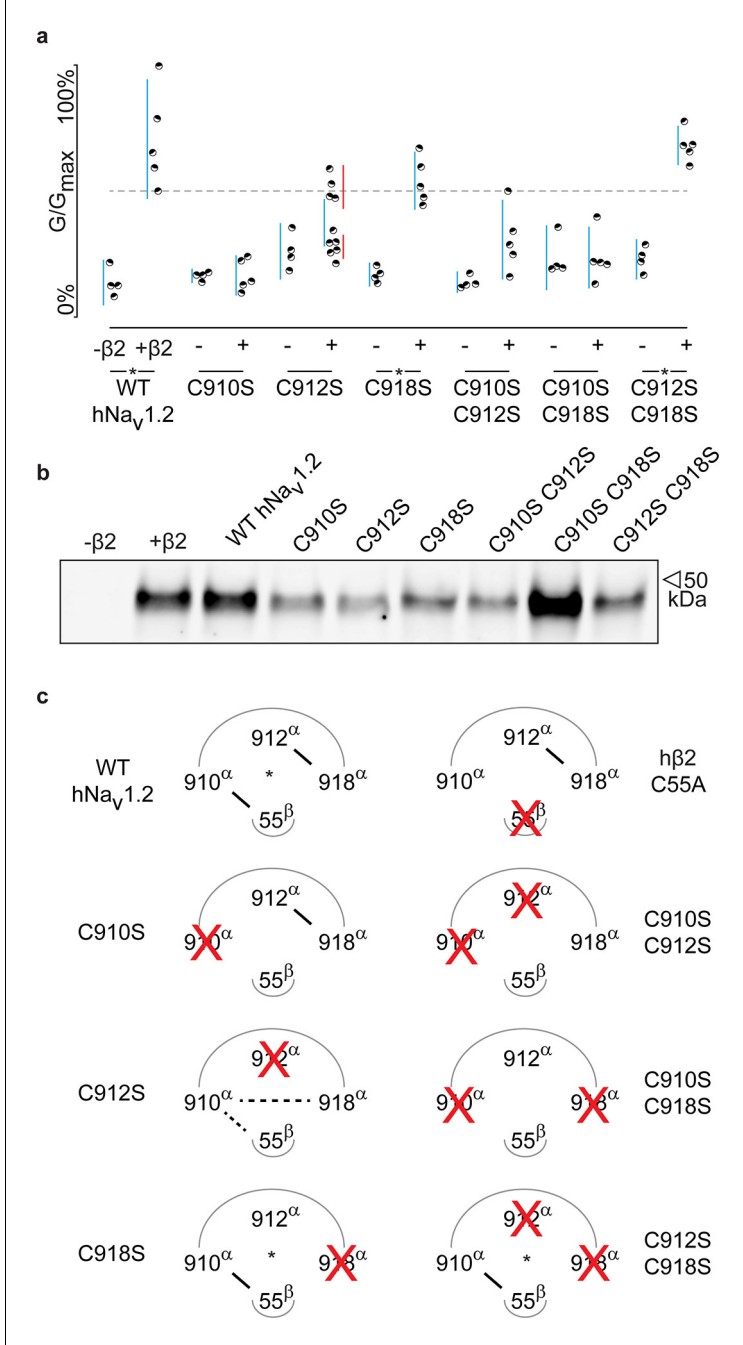

**Figure 4.** h$\beta$2 forms a disulfide bond with $^{910}$Cys in hNa$_v$1.2. (**a**) Dot plot showing degree of hNav1.2 mutant inhibition by 100 nM ProTx-II, with and without h$\beta$2. Black circles represent single oocytes expressing the indicated constructs and currents were measured at peak conductance. Blue bars indicate 95% confidence interval, and the red bars are 95% confidence intervals for both populations in the C912S mutant co-expressed with h$\beta$2. Statistical significance with p.01 is shown by an asterisk. (**b**) Western blot against a myc-tag reveals the presence of h$\beta$2 in whole cell fraction of oocytes co-injected with h$\beta$2 mRNA. (**c**) Schematic depiction of the proposed disulfide arrangement in hNa$_v$1.2 mutants with h$\beta$2. hNa$_v$1.2 cysteines are displayed on the top arc with $\alpha$ (pore-forming subunit), while $^{55}$Cys of h$\beta$2 is on the bottom arc with $\beta$. Black lines represent putative disulfide bonds and dashed lines indicate alternate possibilities. Red X symbolizes a Cys to Ser mutation and an asterisk denotes mutants protected against 100 nM ProTx-II.

The following source data and figure supplements are available for figure 4:

*Figure 4 continued*

**Source data 1.** Table providing values for fits of the data presented in *Figure 4* and *Figure 4—figure supplement 1*.

**Source data 2.** Table providing values for fits of the data presented in *Figure 4—figure supplement 2*.

**Figure supplement 1.** G-V and SSI relationships for hNa$_v$1.2 mutants.

**Figure supplement 2.** Activity of the $\mu$O§-conotoxin GVIIJ is affected by h$\beta$2.

much clearer picture wherein h$\beta$2 only protects against ProTx-II when [910]Cys is intact: the C912S C918S mutant still has lowered toxin susceptibility whereas C910S C912S and C910S C918S are completely inhibited by 100 nM ProTx-II, suggesting that these mutants no longer bind h$\beta$2 (*Figure 4a–b*, *Figure 4—figure supplement 1*). Altogether, these results indicate that [910]Cys is the disulfide bond partner of [55]Cys in h$\beta$2, while [912]Cys and [918]Cys could form an intra-subunit bridge. At first sight, the C912S data which show the split population appear in conflict with this interpretation. However, it is conceivable that losing the [912]Cys- [918]Cys bond results in the formation of a non-native bond between [910]Cys and [918]Cys (*Figure 4c*). Indeed, when [918]Cys is mutated in addition to [912]Cys, [918]Cys again allows toxin protection by h$\beta$2 similar to WT, suggesting it has become available again. The data also indicate that a non-native disulfide bond between [910]Cys and [912]Cys may not occur in the C918S mutant, likely because such a Cys-Val-Cys disulfide bond could be too constrained. Importantly, h$\beta$2 cannot protect the C910S C918S or the C910S C912S mutants from 100 nM ProTx-II, indicating that neither [912]Cys nor [918]Cys can compensate for the loss of [910]Cys (*Figure 4c*).[918]

To biochemically verify that Na$_v$1.2 and $\beta$2 are covalently bound, we expressed the closely related and well-expressing rat variants (~99% sequence identity) in oocytes and immunoprecipitated the rNa$_v$1.2a/r$\beta$2 complex. It is worth noting that WT myc-tagged r$\beta$2 can form higher-order oligomers under non-reducing conditions, which may complicate the interpretation of immunoblots (*Figure 5—figure supplement 1*). However, these oligomers disappear upon removal of [55]Cys, further highlighting the reactivity of this residue. Notwithstanding this phenomenon, probing crude cell lysate uncovers a clear co-migration of myc-tagged r$\beta$2 with rNa$_v$1.2a under non-reducing conditions, since co-expression yields a distinct myc-stained band above the highest r$\beta$2 oligomer (*Figure 5a*). Substitution of either [55]Cys in r$\beta$2 or [910]Cys in rNa$_v$1.2a with Ser results in the loss of this covalent complex. Furthermore, the presence of DTT completely abolishes the r$\beta$2-rNa$_v$1.2a complex as well as the r$\beta$2 oligomers (*Figure 5b*). Thus, these results confirm our toxin experiments and suggest the formation of a disulfide bond between [910]Cys in the DII S5-S6 linker of Na$_v$1.2 and [55]Cys in $\beta$2. In addition to investigating the interaction of rNa$_v$1.2a and r$\beta$2 in crude lysate from injected oocytes, we also pulled down r$\beta$2 with an antibody directed against the C-terminal myc-tag and treated with DTT before loading onto a tris-acetate gel (*Figure 5c*). In this experiment, we observe that WT rNa$_v$1.2a is present only when co-expressed with WT r$\beta$2. Moreover, both [910]Cys within the channel and [55]Cys in r$\beta$2 are required for co-immunoprecipitation, thereby pointing to their role in forming a disulfide bond between both subunits (*Figure 3*, *Figure 4*).

Next, we were curious to learn whether [910]Cys also constitutes an inter-subunit anchoring point for [58]Cys in the $\beta$4-subunit. To examine this notion, we measured ProTx-II susceptibility of rNa$_v$1.2a and its C910S mutant expressed in oocytes without or with the r$\beta$4-subunit (*Figure 6a,b*, *Figure 6—source data 1*). Analogous to h$\beta$2, we observe r$\beta$4 protein production in oocytes (*Figure 6c*) where it is able to influence the degree of rNa$_v$1.2a current inhibition by ProTx-II. In particular, 100 nM ProTx-II reduces rNa$_v$1.2a conductance to ~22% of peak whereas the current remaining in the presence of r$\beta$4 is ~55% of peak conductance. Other gating parameters such as G–V and channel availability relationships are unaffected. In case of the C910S mutant, the presence of r$\beta$4 no longer decreases ProTx-II efficacy (from ~22% current inhibition to ~17%, respectively, *Figure 6—source data 1*), thus illustrating the likely role of rNa$_v$1.2a [910]Cys in forming a disulfide bond with r$\beta$4 [58]Cys. In concert, (co-)immunoprecipitation experiments with rNa$_v$1.2a and r$\beta$4 expressed in oocytes

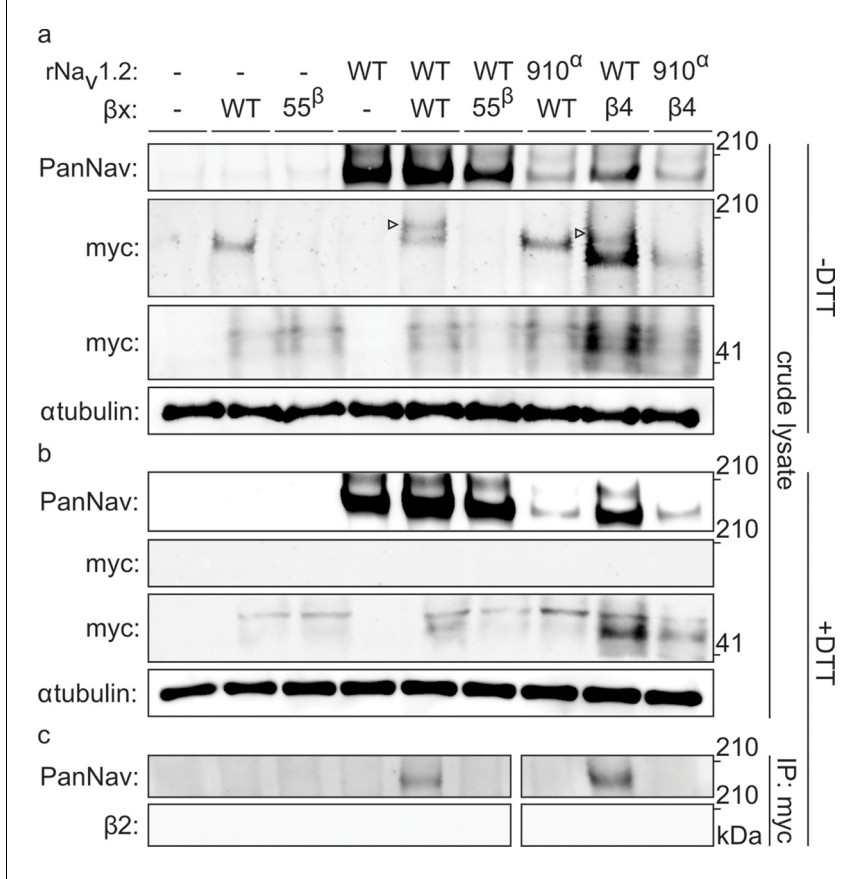

**Figure 5.** Biochemical verification of the β2 and β4 disulfide bond with $^{910}$Cys in Na$_v$1.2. Top rows indicate particular Na$_v$ channel and β-subunit constructs with which oocytes were injected. WT proteins are indicated as such whereas mutants are noted with a residue number; a superscript letter symbolizes in which partner of the complex pair the mutation is found. rβ2 was loaded in each case except in the two rightmost lanes, where rβ4 was used. Labels on left column indicate which antibody was used to immunoblot the associated slice. (**a**) Western blot run under non-reducing conditions reveals a fraction of rβ2 migrating with WT rNa$_v$1.2a but not after selective cysteine substitution. Crude lysate from injected oocytes was run on a protein gel and probed for both the channel and the myc-tag of the β-subunit. The top PanNav slice and bottom myc slice show the expression of the respective proteins in lysates from oocytes injected with the indicated constructs. Even though equal quantities of protein was loaded in each lane (10 μg), the C910S channel mutant expresses less bountifully than the WT channel and as a result shows a weaker signal. The WT β-subunit can form redox-sensitive multimers that migrate at an apparent mass similar to that of the Na$_v$ channel. Mutation of $^{55}$Cys prevents multimer formation and disappearance of the high molecular weight band. Open arrows identify bands representing Na$_v$ channel-bound β-subunit and aid in distinguishing them from the multimeric β-subunit. Substituting $^{910}$Cys within rNa$_v$1.2a with Ser causes a loss of the channel-bound β-subunit band. rβ4 also binds to WT rNav1.2a, as evidenced by a second band, and no longer interacts with the C910S mutant. (**b**) Addition of DTT prevents both binding of rβ2 to the channel and formation of β-subunit multimers. The absence of myc signal at the same apparent weight as rNa$_v$1.2a indicates that binding to the channel is sensitive to reduction. In all cases, α-tubulin was used as a loading control. (**c**) WT rNa$_v$1.2a co-immunoprecipitates with rβ2 and rβ4. The β-subunit was pulled down with an antibody directed against a C-terminal myc-tag and treated with DTT before loading onto the gel. The channel is present only when the WT channel is co-expressed with the WT β-subunit.

The following figure supplement is available for figure 5:

**Figure supplement 1.** Biochemical assessment of rβ2 oligomer formation.

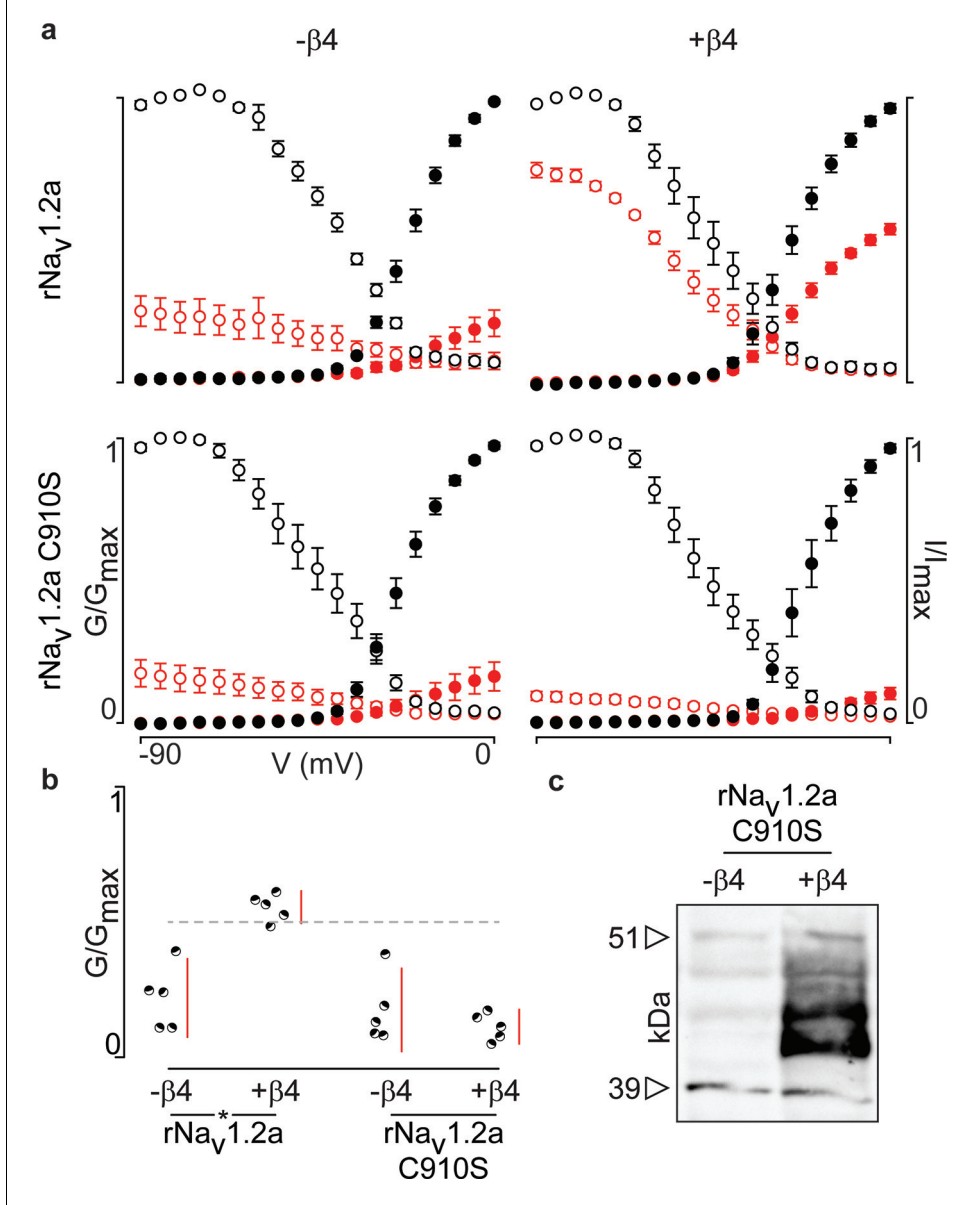

**Figure 6.** Mutation of [910]Cys in rNa$_v$1.2a disrupts r$\beta$4 influence on ProTx-II effect. (**a**) Replacement of [910]Cys with Ser in rNa$_v$1.2a impairs the ability of r$\beta$4 to protect against 100 nM ProTx-II inhibition in a manner similar to that of h$\beta$2. Co-expression of the rNa$_v$1.2a isoform with r$\beta$4 minimizes ProTx-II inhibition compared to the channel expressed alone. Graphs compare G-V and SSI relationships, in filled and open circles, respectively. Black color is used for values before the addition of 100 nM ProTx-II and red after. (**b**) Dot plot comparing the degree of inhibition by 100 nM ProTx-II as a fraction of the pre-toxin peak current. r$\beta$4 confers a high level of a protection against ProTx-II inhibition which is absent in the C910S mutant. (**c**) Western blot directed against the C-terminal myc-tag of r$\beta$4 demonstrating its presence in the oocytes measured.

The following source data is available for figure 6:

**Source data 1.** Table providing values for fits of the data presented in **Figure 6**.

indicate that both partners are covalently bound and that mutating [910]Cys in the channel indeed disrupts the disulfide bond (**Figure 5**).

Aligning the primary sequences of all tetrodotoxin (TTX)-sensitive hNa$_v$ channel isoforms (Na$_v$1.1–1.7) reveals that the cardiac hNa$_v$1.5 channel lacks the cysteine triad while the flanking sequences

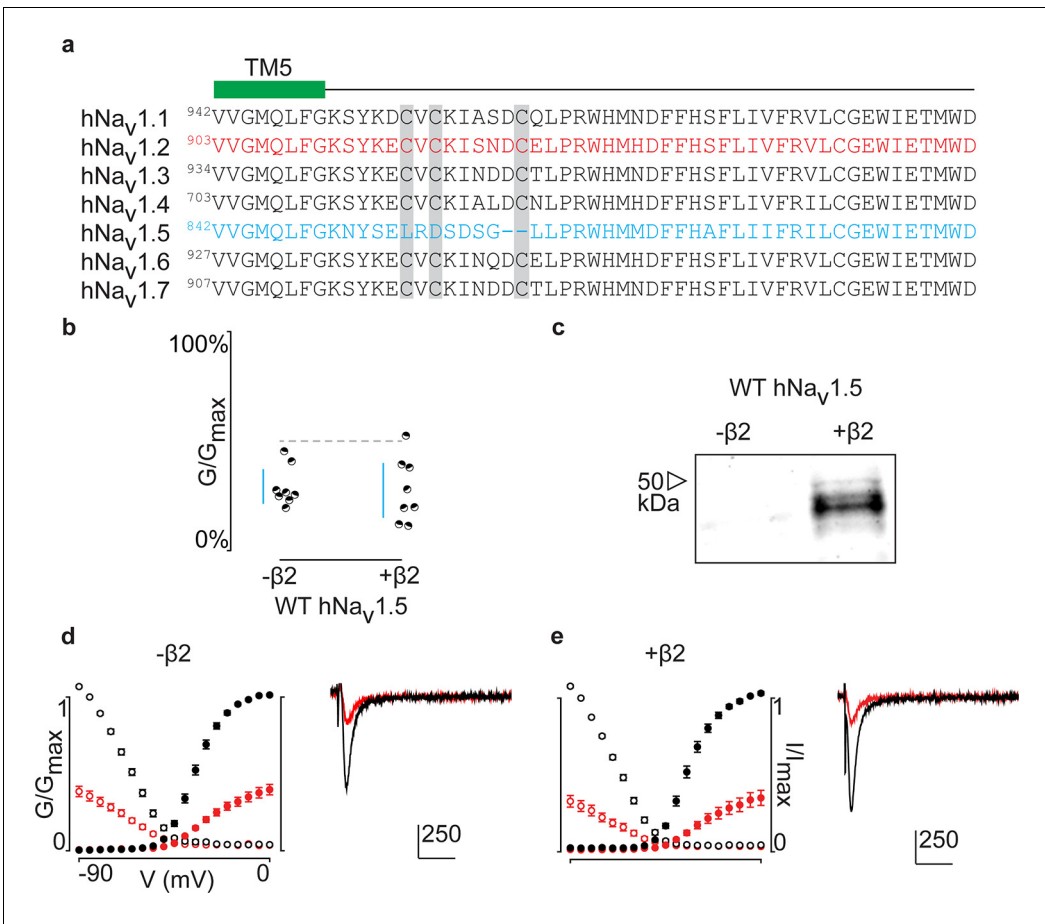

**Figure 7.** ProTx-II inhibits hNa$_v$1.5 in the presence of h$\beta$2. (**a**) Sequence alignment comparing TTX-sensitive hNa$_v$ channels in the beginning of the S5-S6 (SS1) loop. Green bar indicates the C-terminal portion of the DII transmembrane segment 5 (TM5), and gray background highlights the conserved cysteine triad. The hNa$_v$1.2 amino acid sequence is shown in red and hNa$_v$1.5 in blue. Residue number of the N-terminal Val for each channel is superscripted. (**b**) Dot plot showing that hNa$_v$1.5 is not protected against inhibition by 100 nM ProTx-II upon co-expressing h$\beta$2. Black circles are individual oocytes of which sodium currents were measured at peak conductance and blue bars show 95% confidence interval. (**c**) Western blot probing for the C-terminal myc-tag of h$\beta$2 reveals its presence in whole cell oocyte fractions. hNa$_v$1.5 sees no change in 100 nM ProTx-II effect in the presence of h$\beta$2. (**d, e**) G-V (filled circles) and SSI (open circles) relationships for hNa$_v$1.5 with and without h$\beta$2. Black is used for values before ProTx-II addition and red after toxin application. Also shown are representative traces illustrating the lack of effect of h$\beta$2 on hNav1.5 susceptibility to 100 ProTx-II. Horizontal bar indicates 10 ms; vertical bar represents current in nA, with the provided magnitude.
The following source data is available for figure 7:

**Source data 1.** Table providing values for fits of the data presented in **Figure 7**.

are highly conserved (**Figure 7a**). Subsequently, it seems unlikely that hNa$_v$1.5 forms a covalent bond with h$\beta$2, unless it occurs via a distinct site. However, when applying 100 nM ProTx-II to hNa$_v$1.5 without and with the $\beta$-subunit, we observe no effects on gating or protection against the toxin, suggesting that h$\beta$2 may not modulate this channel isoform (**Figure 7b–e**, **Figure 7—source data 1**). It is worth mentioning that hNa$_v$1.8 and hNa$_v$1.9, two TTX-resistant isoforms that are evolutionary related to hNa$_v$1.5, also lack the cysteine triad and may therefore not interact with h$\beta$2 via a disulfide bond.

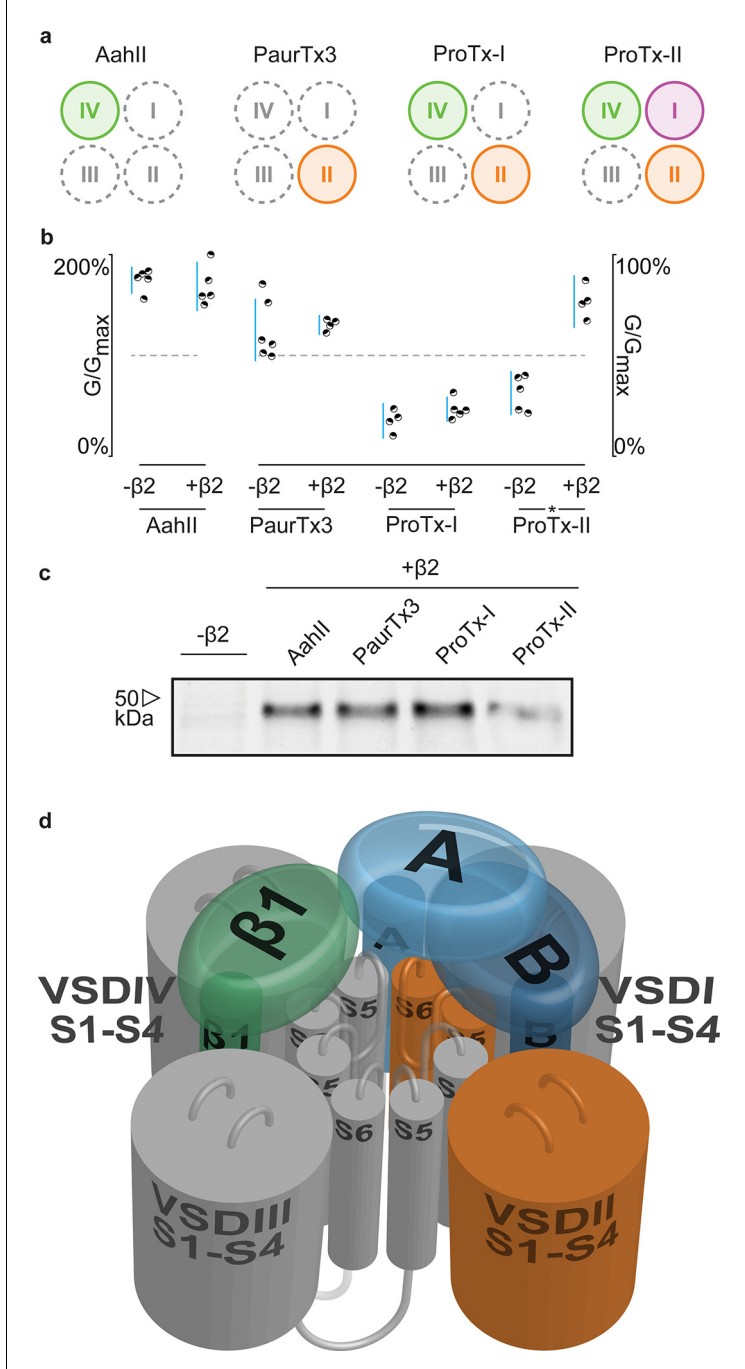

**Figure 8.** h$\beta$2 influences hNa$_v$1.2 VSDI toxin pharmacology. (**a**) Cartoon illustrating the binding site of AaHII, PaurTx3, ProTx-I, and ProTx-II within the VSDs of Na$_v$1.2. Binding to a particular VSD is indicated by coloration. (**b**) Dot plot showing percent of change in peak conductance for hNa$_v$1.2 after treatment by each toxin without or in the presence of h$\beta$2. Blue bars represent 95% confidence interval and statistical significance (p<0.01) is noted by an asterisk. **c**) Western blot probing for the C-terminal myc-tag of h$\beta$2, showing its presence in each experimental condition except the negative control. (**d**) Shown is a Na$_v$ channel illustration consisting of the four VSDs I-IV (S1-S4) and the corresponding pore-forming regions (S5-S6) in grey (DI, III, and IV) or orange (DII) in a clockwise orientation which fits with the data reported here. Such an orientation places a S1-S4 voltage sensor of one domain adjacent to the S5-S6 region of the next. A/B (shades of blue) depicts a putative location of h$\beta$2 in the complex where the subunit can interact with the S5-S6 pore region of DII as well as occlude the ProTx-II binding site on VSDI. In contrast, h$\beta$1 (green) may position itself between VSDIV and VSDIII where it can interact with the S5-S6 region of DI and influence VSDIV movement to alter channel fast inactivation.

*Figure 8 continued on next page*

*Figure 8 continued*

The following source data and figure supplement are available for figure 8:

**Source data 1.** Table providing values for fits of the data presented in *Figure 8* and *Figure 8—figure supplement 1*.

**Figure supplement 1.** Activity of spider and scorpion toxins on hNa$_v$1.2.

## Positioning β2 in relation to Na$_v$1.2

Having identified [910]Cys within the DII S5-S6 loop of hNa$_v$1.2 and [55]Cys within hβ2 as inter-subunit disulfide bond partners, we sought to find evidence for a potential locus of the β-subunit in relation to the channel. Although hβ2 seems to be anchored to the DII pore forming region, ProTx-II was previously shown to target VSDs I, II, and IV (*Bosmans et al., 2008*), suggesting that the protective effect of hβ2 is through occlusion of the ProTx-II binding site in one or more of these regions. In order to determine which one, we assembled a set of animal toxins that, together, interact with VSDI, II, and IV in Na$_v$1.2 (*Bosmans et al., 2008*), and tested whether or not their function is influenced by hβ2 (*Figure 8a*). A previous study in which rNa$_v$1.2a S3b-S4 paddle loops from each domain (I-IV) were transplanted into a homotetrameric K$_v$ channel to identify the VSDs with which toxins interact (*Bosmans et al., 2008*), found that the tarantula toxin PaurTx3 and scorpion toxin AahII (*Martin et al., 1987*) exclusively target VSDII and VSDIV, respectively. In addition, the tarantula toxins ProTx-I and ProTx-II both interact with the voltage sensor in DII and DIV, whereas ProTx-II also binds to DI with high affinity. Here, we tested these four toxins on hNa$_v$1.2 without and with hβ2 to determine if the presence of the subunit impacts toxin function. Aside from ProTx-II, we observe no significant difference in PaurTx3, AahII, or ProTx-I effect which suggests that hβ2 does not impede binding of these toxins to VSDII and VSDIV (*Figure 8b–c*, *Figure 8—source data 1* and *Figure 8—figure supplement 1*). Therefore, we speculate that hβ2 primarily influences VSDI and as such, is located near this region (*Figure 8d*). Since µO§-conotoxin GVIIJ function is also influenced by hβ2 (*Gajewiak et al., 2014*), it will be interesting to investigate a possible VSDI binding site for this toxin. While our data are limited by the absence of VSDI- and VSDIII-specific toxins and do not exclude the possibility of non-overlapping binding sites for a particular toxin and hβ2 on the same VSD, or of direct competition for binding to the DII S5-S6 loop, they may yet prove valuable as insights into hβ2 function accrue.

## Discussion

The goal of this study was to explore the interaction of β2 with Na$_v$1.2 and identify anchoring residues in both partners which will help orient functional motifs within their extracellular domains. Although we obtained the first crystal structure of the hβ4 extracellular region (*Gilchrist et al., 2013*), the second reported structure of the corresponding hβ3 domain (*Namadurai et al., 2014*) highlights the unique character of each β-subunit isoform. Since the ability to compare the distinct organizational features of β-subunits will provide valuable information as to their difference in action, we now obtained detailed structural information for the extracellular domain of hβ2 at a resolution of 1.35Å (*Figure 1a–b*, *Figure 1—source data 1*). We identified a flexible loop formed by [72]Cys and [75]Cys that is a unique feature among β-subunits but with a function that has yet to be elucidated (*Figure 1c*, *Figure 2*, and *Figure 3*). Moreover, hβ2 contains a Cys at position 55 that, when mutated, disrupts the influence of hβ2 on hNa$_v$1.2 toxin pharmacology (*Figure 3b–d*). Next, we combined mutagenesis and biochemical studies with spider and scorpion toxins that target specific VSDs within Na$_v$1.2 (*Gilchrist et al., 2014*; *Bosmans et al., 2008*) to probe the interaction with hβ2. As a result, we found that [55]Cys forms a distinct disulfide bond with [910]Cys located in the domain II S5-S6 loop of hNa$_v$1.2, thereby revealing a 1:1 stoichiometry (*Figure 4a–b*, *Figure 4—source data 1* and *Figure 4—figure supplement 1*). We also exploited this toxin-reporter approach to investigate the possibility of intra-subunit disulfide bond formations between [910]Cys, [912]Cys, and [918]Cys in hNa$_v$1.2, three reactive residues that are in close vicinity to each other. The outcome of these experiments indicates that [912]Cys and [918]Cys form an intra-subunit bridge in WT hNa$_v$1.2 (*Figure 4*). When

mutating $^{918}$Cys, hβ2 still interacts with $^{910}$Cys whereas substituting $^{912}$Cys suggests the possibility for bond formation between $^{910}$Cys and $^{918}$Cys (*Figure 4c*).

Interestingly, hNa$_v$1.5 lacks these three cysteines (*Figure 7a*) and as a result, ProTx-II is equipotent without or in the presence of hβ2 (*Figure 7b–e*, and  *Figure 7—source data 1*) This outcome provides evidence that this particular β-subunit may not modulate hNa$_v$1.5 in heterologous systems or that binding occurs through a different mechanism in native tissues. In concert, immunocytochemical studies in the heart and electrophysiological measurements in mammalian cell lines or oocytes disagree on whether hβ2 can modulate hNa$_v$1.5 function (*Johnson and Bennett, 2006*; *Zimmer and Benndorf, 2002*, *2007*; *Dhar Malhotra et al., 2001*; *Maier et al., 2004*). As opposed to native cardiomyocytes, hNa$_v$1.5 may not associate with hβ2 upon heterologous expression, a hypothesis that is supported by co-localization experiments in HEK293 cells where hNa$_v$1.5 and hβ2 were mainly found in the endoplasmatic reticulum or plasma membrane, respectively (*Zimmer et al., 2002*).

So far, none of the hβ2 mutations implicated in disorders have been found close to $^{55}$Cys, as they are located either in the signal peptide or near the transmembrane region (*Li et al., 2013*; *Medeiros-Domingo et al., 2007*; *Tan et al., 2010*; *Baum et al., 2014*; *Watanabe et al., 2009*). In contrast, amino acid substitutions in the DII S5-S6 pore loop of particular Na$_v$ channel variants are linked to diseases. For example, more than 30 mutations have been identified within this region in hNa$_v$1.1, several of which relate to Dravet syndrome (*Claes et al., 2009*). This includes C927F (corresponding to $^{918}$Cys in hNa$_v$1.2), and other variants which introduce an additional Cys that may interfere with local disulfide bond pattern formation. Therefore, it is conceivable that DII S5-S6 linker mutations may affect channel interactions with hβ2 or hβ4.

Collectively, our results uncover the disulfide link between hβ2 and hNa$_v$1.2 which opens up the possibility to assign a potential orientation of this subunit in relation to the channel and provide an experimental basis for future docking efforts (*Figure 8d*). hβ2 may position itself in the gaps between VSDs where it can interact with a voltage sensor as well as anchor to a pore-forming region. One important observation from bacterial Na$_v$ channel crystal structures (*Payandeh et al., 2012*; *Payandeh et al., 2011*; *Shaya et al., 2014*; *Zhang et al., 2012*) is the domain-swapped architecture of the channel in which the S1-S4 VSD within one subunit is located adjacent to the S5-S6 segments of the next subunit. Such a structural clockwise arrangement has also been consistently observed in prokaryotic and mammalian K$_v$ channels (*Long et al., 2007*; *Jiang et al., 2003*). Since hNa$_v$ channels may be organized in a similar fashion, the determinants of hβ2-subunit sensitivity can be located in multiple Na$_v$ channel domains. For example, we found that hβ2 binds to $^{910}$Cys in the S5-S6 loop of DII and influences ProTx-II interaction with VSDI. Although it is challenging to pinpoint its precise location, our data suggest that hβ2 is positioned in the cleft between VSDIV and VSDI or VSDI and VSDII where it is presented with an opportunity to interact with both determinants to influence ProTx-II action (*Figure 8d*). In contrast, the consistent picture emerging from the literature is that Na$_v$ channel fast inactivation and voltage-dependence of activation changes substantially when co-expressed with the β1-subunit in heterologous systems (*Chen and Cannon, 1995*; *McCormick et al., 1998*; *Qu et al., 1999*; *Isom et al., 1995b*). Given the pivotal role of VSDIV in channel fast inactivation (*Bosmans et al., 2008*; *Capes et al., 2013*; *Chanda and Bezanilla, 2002*; *Horn et al., 2000*; *Sheets et al., 1999*), β1 may be positioned close to VSDIV where it can also interact with the extracellular S5-S6 pore-loop within DI (*Figure 8d*), a presumed interacting region (*Qu et al., 1999*; *Makita et al., 1996*). Altogether, this model suggests that particular Na$_v$ channel isoforms can interact simultaneously with a β1- and β2-subunit (*Calhoun and Isom, 2014*). Future experiments will further investigate the possible locations of β3 and β4 in this model as well as allow the incorporation of mutational effects which is essential to uncover the contribution of β-subunit mutations to human disorders or contribute to small molecule screening efforts geared towards disrupting or facilitating subunit interactions.

## Materials and methods

### Production of the hβ2 extracellular domain

Human β2 (residues 30–153) was cloned into a modified pET28 vector (pET28HMT) (*Van Petegem et al., 2004*). Mutations (C55A and C55/72/75A) were introduced using the Quikchange kit from Agilent Technologies (USA) according to the manufacturer's instructions. Proteins were expressed at

18°C in *E. coli* Rosetta (DE3) pLacl strains (Novagen, USA), induced at an OD600 of ~0.6 with 0.4 mM IPTG, and grown overnight prior to harvesting. Cells were lysed via sonication in buffer A (250 mM KCl and 10 mM HEPES at pH 7.4), supplemented with 25 µg/ml DNaseI and 25 µg/ml lysozyme. After centrifugation, the supernatant was applied to a PorosMC column (Tosoh Biosep, USA), washed with buffer A plus 10 mM imidazole, and eluted with buffer B (250 mM KCl plus 500 mM imidazole pH 7.4). The protein was dialyzed overnight against buffer A and cleaved simultaneously with recombinant TEV protease. Next, the samples were run on another PorosMC column in buffer A, and the flowthrough was collected and dialyzed against buffer C (10 mM KCl plus 10 HEPES at pH 7.4), applied to a HiloadQ column (GE Healthcare, USA), and eluted with a gradient from 0% to 30% buffer D (1 M KCl plus 10 mM HEPES at pH 7.4). Finally, the samples were run on a Superdex200 (GE Healthcare, USA) gel filtration column in buffer A. The protein samples were exchanged to 50 mM KCl plus 10 mM HEPES (pH 7.4), concentrated to ~5 mg/ml using Amicon concentrators (3K MWCO; Millipore USA), and stored at −80°C.

## Crystallization, data collection, and structure solution

Crystals were grown using the hanging-drop method at 4°C. Both C55A and C55/72/75A were crystallized in 0.1 M Tris (pH 8), 15–20% (w/v) PEG 6000. Crystals were flash-frozen after transfer to the same solution supplemented with 30% glycerol. The data used to solve the final structures were collected at the Canadian Light Source (Saskatoon) beamline 08ID-1 and datasets were processed using XDS (*Kabsch, 2010*) and HKL3000 (*Minor et al., 2006*). A search model was created by using only β-strands from PDB 4 MZ2, and with all side chains truncated to alanine. Molecular replacement was performed using Phaser (*McCoy et al., 2007*), yielding poor initial phases which were improved via autobuilding in ARP/wARP (*Langer et al., 2008*). The model was completed by successive rounds of manual model building in COOT (*Emsley et al., 2010*) and refinement using Phenix (*Adams et al., 2010*). A simulated annealing composite omit map was calculated with CNS (*Brünger et al., 1998*) to verify the absence of residual model bias. [85]Met was found to be in a disallowed region of the Ramachandran plot in both structures. All structure figures were prepared using PYMOL (DeLano Scientific, San Carlos, USA).

## Accession codes

Coordinates and structure factors have been deposited in the Protein Data Bank under accession codes 5FEB (C55A) and 5FDY (C55/72/75A).

## Toxin acquisition and purification

ProTx-I and ProTx-II were acquired from Peptides International (USA), PaurTx3 from Alomone Labs (Israel). AaHII from *Androctonus australis* hector venom was purified as described (*Martin et al., 1987*). Toxins were kept at -20°C and aliquots were dissolved in appropriate solutions containing 0.1% BSA.

## Two-electrode voltage-clamp recording from *Xenopus* oocytes

The DNA sequence of hNa$_v$1.2 (NM_021007.2), rNa$_v$1.2a (NM_012647.1), rβ4 (NM_001008880) and hβ2 (NM_004588.4) (acquired from Origene, USA), as well as their mutants was confirmed by automated DNA sequencing and cRNA was synthesized using T7 polymerase (mMessage mMachine kit, Ambion) after linearizing the DNA with appropriate restriction enzymes. Channels were expressed in *Xenopus* oocytes together with a β-subunit (1:5 molar ratio) and studied following 1–2 days incubation after cRNA injection (incubated at 17°C in 96 mM NaCl, 2 mM KCl, 5 mM HEPES, 1 mM MgCl$_2$ and 1.8 mM CaCl$_2$, 50 µg/ml gentamycin, pH 7.6 with NaOH) using two-electrode voltage-clamp recording techniques (OC-725C, Warner Instruments) with a 150 µl recording chamber. The data were filtered at 4 kHz and digitized at 20 kHz using pClamp 10 software (Molecular Devices, USA). Microelectrode resistances were 0.5–1 MΩ when filled with 3 M KCl. The external recording solution contained 100 mM NaCl, 5 mM HEPES, 1 mM MgCl$_2$ and 1.8 mM CaCl$_2$, pH 7.6 with NaOH. The experiments were performed at room temperature (~22°C) and leak and background conductances, identified by blocking the channel with tetrodotoxin (Alomone Labs, Israel), have been subtracted for all Na$_v$ channel currents. All chemicals used were obtained from Sigma-Aldrich (USA) unless indicated otherwise.

## Analysis of channel activity and toxin–channel interactions

Voltage–activation relationships were obtained by measuring steady-state currents and calculating conductance (G), anitted to the data according to: $G/G_{\max} = \left(1 + e^{-zF(V - V_{1/2})/RT}\right)^{-1}$ where G/$G_{\max}$ is the normalized conductance, $z$ is the equivalent charge, $V_{1/2}$ is the half-activation voltage, $F$ is Faraday's constant, R is the gas constant and T is temperature in Kelvin. Occupancy of closed or resting channels by ProTx-II and other toxins was examined using negative holding voltages where open probability was very low, and the fraction of uninhibited channels (Fu) was estimated using depolarizations that are too weak to open toxin-bound channels, as described previously. After addition of the toxin to the recording chamber, the equilibration between the toxin and the channel was monitored using weak depolarizations typically elicited at 5 s intervals. Off-line data analysis was performed using Clampfit 10 (Molecular Devices, USA), and Origin 8 (Originlab, USA).

## Qualitative biochemical assessment of hβ2 production in *Xenopus* oocytes

After each electrophysiological experiment, oocytes expressing hNa$_v$1.2, hβ2, and the described mutants were washed with ND100 and incubated with 0.5 mg/ml Sulfo-NHS-LC-biotin (Pierce, USA) for 30 min. Oocytes were thoroughly washed again in ND100 before lysis (by pipetting up and down) in 20 µl/oocyte buffer H (1% Triton X-100, 100 mM NaCl, 20 mM Tris-HCl, pH 7.4) plus protease inhibitors (Clontech, USA). All subsequent steps were performed at 4°C. Lysates were gently shaken for 15 min after which they were centrifuged at 16,200x*g* for 3 min. The pellet was discarded and the supernatant (SN) transferred to a fresh 1.l Eppendorf tube. 40 µl of SN was stored at -80°C for later use as the whole cell protein aliquot. 200 µl of hydrophilic streptavidin magnetic beads (New England Biolabs, USA) were then added and the sample shaken gently at 4°C overnight. Beads were washed 6 times with buffer H and resuspended in 40 µl buffer H, after which the biotinylated protein was dissociated from the beads through the addition of 1X LDS loading buffer plus reducing agent (10% 2-ME, 50 mM DTT final conc.) and boiling at 95°C for 5 min to generate the surface protein fraction. All samples were appropriately diluted in buffer H to give equal protein concentrations, as measured by a BCA assay (Pierce, USA). 10 µg of the SN was run on a 10% Tris-Glycine Novex Mini-Gel (Thermo Fisher Scientific, USA) with Tris-Glycine running buffer and analyzed by Western analysis. Nitrocellulose membranes were probed overnight at 4°C with 1:1000 mouse anti-myc antibody as primary (Cell Signaling Technologies, USA) and for 45 min at room temperature with 1:10000 goat anti-mouse HRP-conjugated antibody as secondary (Thermo-Fisher Scientific, USA). Membranes were incubated for 5 min with an enhanced chemiluminescent substrate before imaging.

## Immunoprecipitation of the Na$_v$1.2/β2 complex

*Xenopus* oocytes were injected with 5 ng RNA of rβ2, rNa$_v$1.2, or both (~1:5 molar ratio) and incubated for 48 hr at 17°C. 30 oocytes were lysed for each condition, using 20 µl lysis buffer (1x PBS, 1% DDM, 10% glycerol, 1 mM EDTA, 1X protease inhibitor cocktail [Clontech, USA]) per oocyte, and homogenized by passing through a 25-gauge syringe (adapted from [*Yu, 2012*]). The lysate was rotated for 1 hr at 4°C in a 1.5 ml microfuge tube, after which it was spun for 30 min at 20,000x*g* at 4°C. Subsequent steps were performed at 4°C or on ice and during all rotations the tube was sealed with parafilm to prevent leakage. A fresh pipette tip was gently swirled in the supernatant to remove the bulk of the white goop by adhering to the tip and supernatant was transferred to a new tube, taking care not to disturb the pellet. The new tube was spun 3 min at 20,000x*g* and again was swirled with a fresh pipette tip to remove the white goop, after which the clear supernatant was transferred to a new tube. Protein concentration was assayed using a BCA protein concentration kit (Pierce, USA). 150 µg protein was brought to 150 µl in lysis buffer and 1.5 µg (1.5 µl of 1 mg/ml) anti-myc mouse antibody (Thermo-Fisher Scientific, USA) was added. The tube was rotated overnight at 4°C after which 30 µl protein G-coated magnetic Dynabeads (Thermo-Fisher Scientific, USA) were added and rotated for 4 hr at 4°C. The tube was spun for 1 min at 20,000x*g* and placed on a magnetic rack for 1 min to collect the beads. Supernatant was removed and stored for optimization and troubleshooting purposes, as were subsequent washes. The beads were washed with 200 µl wash buffer (1x PBS, 0.4% DDM, 10% glycerol, 1 mM EDTA) by pipette mixing and then magnetized to collect beads. The wash process was repeated three times. After the third wash, beads were

suspended in 100 μl wash buffer, transferred to another tube, spun for 1 min at 20,000x$g$, and magnetized to collect beads. The tube was then spun again for 1 min at 20,000x$g$ and magnetized, followed by removal of the last residual fluid. Protein was eluted in 30 μl elution buffer (50 mM glycine, pH 2.8) by incubating for 3 min at room temperature, followed by magnetization and transfer to the final tube. For Western blotting of the immunoprecipitate, 17 μl of eluate was combined with 2.5 μl 10X reducing agent (Thermo-Fisher Scientific, USA) and 6.5 μl 4X LDS sample loading buffer (Thermo-Fisher Scientific, USA), heated for 10 min at 37°C, then loaded onto a 1.0 mm 12-well 3–8% Novex Tris-Acetate pre-cast gel (Thermo-Fisher Scientific, USA). A rabbit anti-PanNav antibody (Alomone labs, Israel) and rabbit anti-hβ2 antibody (Cell Signaling Technologies, USA) were chosen to avoid cross-reaction of the Western blot secondary antibody against the mouse antibody used for immunoprecipitation. For Western blotting, the primary antibodies were used at 1:200 and 1:1000 dilutions, respectively, applied for 1 hr at room temperature and incubated for 45 min at room temperature with a 1:10,000 goat anti-rabbit HRP-conjugated secondary antibody (Thermo-Fisher Scientific, USA).

## Acknowledgements

We would like to thank Marie-France Martin-Eauclaire and Pierre Bougis (University of Marseille, France) for sharing AaHII; Al Goldin (UCIrvine, USA) for sharing rNa$_v$1.2a; YaWen Lu and Elizabeth Calzada (Johns Hopkins University – School of Medicine, USA) for assistance with biochemistry; Michael Pennington (Peptides International, USA) for sharing ProTx-II used in the ITC experiment; and Baldomero Olivera (University of Utah – Salt Lake City, USA) for sharing GVIIJ. We also wish to thank the staff of the Advanced Photon Source (Chicago, USA) GM/CA-CAT beamline 23-ID-D, the Stanford Synchrotron Radiation Lightsource (Menlo Park, USA), and the Canadian Light Source (Saskatoon, SK, Canada), which is supported by the Natural Sciences and Engineering Research Council of Canada, the National Research Council Canada, the Canadian Institutes of Health Research (CIHR), the Province of Saskatchewan, Western Economic Diversification Canada, and the University of Saskatchewan. Parts of this work are supported by the National Institute of Neurological Disorders And Stroke (NINDS) of the National Institutes of Health (NIH) under Award Number 1R01NS091352 to FB, a Human Frontier Science Program grant to FB and FVP (RGY0064/2013), and a CIHR grant to FVP (MOP-119404). SD is supported by a postdoctoral fellowship of the Heart and Stroke Foundation of Canada and JG by the NINDS/NIH under Award Number F31NS084646. The content is solely the responsibility of the authors and does not necessarily represent the official views of the NIH.

## Additional information

### Funding

| Funder | Grant reference number | Author |
| --- | --- | --- |
| National Institutes of Health | 1R01NS091352 | Frank Bosmans |
| Canadian Institutes of Health Research | MOP-119404 | Filip Van Petegem |
| Human Frontier Science Program | RGY0064/2013 | Frank Bosmans Filip Van Petegem |
| Heart and Stroke Foundation of Canada | | Samir Das |
| National Institutes of Health | F31NS084646 | John Gilchrist |

The funders had no role in study design, data collection and interpretation, or the decision to submit the work for publication.

### Author contributions

SD, JG, FB, FVP, Conception and design, Acquisition of data, Analysis and interpretation of data, Drafting or revising the article

## Ethics

Animal experimentation: This study was performed in strict accordance with the recommendations in the Guide for the Care and Use of Laboratory Animals of the National Institutes of Health. All of the animals were handled according to approved institutional animal care and use committee (IACUC) protocols under protocol number AM15M56.

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
