## [Decision Letter]

Thank you for submitting your work entitled "Binary architecture of the Na_v_1.2-β2 signaling complex" for peer review at *eLife*. Your submission has been favorably evaluated by Richard Aldrich (Senior editor), a Reviewing editor, and two reviewers.

The reviewers have discussed the reviews with one another and the Reviewing editor has drafted this decision to help you prepare a revised submission.

Summary:

Das et al., present a structure of a truncated hβ2 subunit structure at ~1.4A and provide functional evidence that the β2 subunit is covalently attached to the α subunit. The structure is of residues 30-153, with a Cys mutation (C55A) to facilitate crystallization. hβ2 has an Ig-like fold, consisting of eleven β-strands and three 310 helices. This domain contains four additional cysteines that are arranged in two bonds, the first intra-subunit bond is conserved among all four β-subunit isoforms and is mediated by ^50^Cys and ^127^Cys buried where it links two opposing faces of the protein. The second intrasubunit bond is in a loop that spans residues 70-77 and connects strand β5 to β6 via ^72^Cys ^75^Cys. This loop is a unique feature of hβ2 since corresponding cysteines are absent in β1, β3, and β4.

In functional studies in which the hβ2 subunit is expressed in oocytes, the authors argue that the Na_v_1.2 alpha subunit is bound by a disulfide link between C55 of the β2 subunit and C910 of the Na_v_1.2 alpha subunit.

Essential revisions:

The reviewers were generally positive about publication, but have two major points to address before resubmission. Specifically:

1) In order to show crosslinking convincingly the authors need to show biochemically that the α and β subunits are covalently bound (by an immunoblot running at the combined weight of the proteins for example), and that this is reducible by DTT and interrupted selectively by mutation of the cysteines in question. The function alone only shows that both Cys are involved in the toxin effect, it is plausible that they both interact individually with the toxin and not each other.

2) All the immunoblots shown need to show loading controls. Quantification of the membrane fraction would substantially strengthen the argument.

---

## [Author Response]

*Essential revisions: The reviewers were generally positive about publication, but have two major points to address before resubmission. Specifically: 1) In order to show crosslinking convincingly the authors need to show biochemically that the α and β subunits are covalently bound (by an immunoblot running at the combined weight of the proteins for example), and that this is reducible by DTT and interrupted selectively by mutation of the cysteines in question. The function alone only shows that both Cys are involved in the toxin effect, it is plausible that they both interact individually with the toxin and not each other.*

We agree with the reviewers that additional biochemical data would strengthen the interpretation of our results with ProTx-II. When we started these experiments however, it quickly became clear that obtaining biochemical evidence of a Nav channel:β-subunit complex is far from trivial. We experimented with a range of conditions (published by others and non-published) but were unable to generate satisfying data within the two-month revision period. Nonetheless, we persisted in our efforts and in the two months thereafter, we managed to clearly show that rNa_v_1.2a and rβ2 form a covalent bond that can be disrupted by mutating _910_Cys and _55_Cys, respectively (an extensive description of the methods has been added to the manuscript). Moreover, we extended these experiments to include rβ4 and found that this subunit can also form a disulfide bond with ^_910_^Cys in rNa_v_1.2a. Finally, we noticed that β2 can form multimers that disappear under reducing conditions or after mutating the reactive ^_55_^Cys to Ser. To our knowledge, this potentially complicating factor in interpreting immunoblots has not yet been reported. As such, the manuscript has been substantially revised to include these data:

*“*To biochemically verify that Nav1.2 and β2 are covalently bound, we expressed the closely related and well-expressing rat variants (~99% sequence identity) in oocytes and immunoprecipitated the rNa_v_1.2a/rβ2 complex. […] In this experiment, we observe that WT rNa_v_1.2a is present only when co-expressed with WT rβ2. Moreover, both ^_910_^Cys within the channel and ^_55_^Cys in rβ2 are required for co-immunoprecipitation, thereby pointing to their role in forming a disulfide bond between both subunits (Figure 3, Figure 4).”

“(Co-)immunoprecipitation experiments with rNa_v_1.2a and rβ4 expressed in oocytes reveal that both partners are covalently bound and that mutating ^_910_^Cys in the channel or ^_58_^Cys in the β-subunit indeed disrupt the disulfide bond (Figure 5).”

2) All the immunoblots shown need to show loading controls. Quantification of the membrane fraction would substantially strengthen the argument.

We certainly agree with the reviewers that immunoblots should show loading controls and have therefore added an α-tubulin control to the newly added experiments described above (see also Figure 5). As far as the other immunoblots in the manuscript are concerned, adding a loading control to those experiments would require doing every experiment again, including the electrophysiological tests to which the blots relate. Since these blots 1) clearly show a qualitative presence of β-subunits in the membrane (which goes beyond most of the reports found in the literature); and 2) result from an equal amount of protein loaded onto the gel each time (10μg), we feel that these immunoblots present an added value in their current form.